# Development of Mode-Switchable Touch Sensor Using MWCNT Composite Conductive Nonwoven Fabric

**DOI:** 10.3390/polym14081545

**Published:** 2022-04-11

**Authors:** Seong Jin Jang, Minhee Kim, Jee Young Lim, Young Ki Park, Jae Hoon Ko

**Affiliations:** 1Korea Institute of Industrial Technology, 143, Hanggaulro, Sangnok-gu, Ansan-si 15588, Korea; anyholic@kitech.re.kr (S.J.J.); specialg@kitech.re.kr (J.Y.L.); parkyk@kitech.re.kr (Y.K.P.); 2Chokwang Paint, 148, 5 Beon-gil, Samdeong-ro, Sasang-gu, Busan-si 46909, Korea; chatelier@ckpc.co.kr; 3Department of Fiber System Engineering, Dankook University, Yongin 16890, Korea

**Keywords:** MWCNT composite conductive nonwoven, textile-based piezo sensor, wearable devices

## Abstract

Among the various wearable electronic devices, textile-based piezo sensors have emerged as the most attractive sensors for practical application. In this study, a conductive nonwoven fabric is fabricated to develop a textile-based piezo sensor. This high-performance fabric is fabricated by depositing multiwalled carbon nanotubes (MWCNTs) on cellulose nonwoven composites with carbon fibers (CNwCa) through a spray process to assign conductivity, followed by electrospinning thermoplastic polyurethane (TPU) on the MWCNT-coated CNwCa to improve surface durability. Each component is optimized through experiments to control the electrical and physical characteristics of the conductive nonwoven fabric. The static and dynamic piezoresistive properties of the fabricated MWCNT composite conductive nonwoven are measured using a source meter and the fabricated sensor driving circuitry. In addition, a prototype bag with a touch sensor is developed using the fabricated conductive nonwoven fabric and its touchpad function is demonstrated using an Android application. The operation as a mode-switchable touch sensor was experimentally verified by inserting the sensor into a bag so that it can be used without direct manipulation on a mobile device. The findings of this study suggest that the developed flexible textile-based conductive nonwoven fabric can be effectively used in wearable devices with piezoresistive sensors.

## 1. Introduction

Wearable electronic devices have attracted significant attention in recent decades [1,2,3,4,5,6,7,8,9,10]. In various types of wearable devices such as smartwatches, fitness trackers, smart clothing, and wearable medical devices [11,12,13], wearable technology has been extensively applied. Wearable electronic devices are required to be lightweight, flexible, breathable, durable, washable, and economically meet practical and esthetic needs, with maximum freedom of movement and maximum comfort. For textiles, in particular, which include several advantages such as negligible physical rejection and ease of washing, there are no limits to the designs that can be created, and they can be manufactured using conventional production equipment. As a representative example, in Levi’s Trucker jacket, developed by Google’s Jacquard Project, a wearable electronic device is introduced in a denim jacket and operated with high recognition accuracy through intuitive gestures such as “brush-in” and “brush-out” [14,15]. Due to the superiority of textiles, textile-based sensors have been and will continue to be selected as a major component of wearable devices [7].

The conductive substrates employed in flexible piezo sensors have been fabricated by applying materials such as carbon nanotubes (CNTs), graphene, and conductive polymers on the surface of flexible materials through the spray, padding, and electrospinning processes [16,17,18,19,20,21,22]. Carbon nanotubes have electrical connectivity due to their high surface area and aspect ratio, and superior mechanical properties. Graphene has a larger surface area than SWCNTs, excellent mechanical properties, and the ability to adjust the electronic bandgap [23,24]. These carbon-based nanoparticles are being applied in various fields based on their remarkable physicochemical properties. Due to the biodegradability and low toxicity of carbon nanoparticles, they have received attention as potential applications in fields such as bio-imaging and gene therapy [25,26,27,28,29]. Although metal-based flexible conductive substrates are available, their electrical properties become unstable due to oxidation when used for a long duration. Recently, there have been several attempts to use CNTs as the raw material for conductive substrates [4,5,6,7,8,9,21,22,30,31,32] or use the electrospinning process for fabricating a conductive substrate in order to realize ultralight, ultrathin, and flexible sensors [16,17,18,33,34,35]. However, a conductive substrate fabricated using the spray process is fragile because of the lack of durability of the conductive layer, and it becomes stiff and thick when a considerable quantity of binder is used to impart durability. In addition, the fabrication of conductive substrates through the electrospinning process is economically disadvantageous because of the high content of conductive material used to impart high conductivity [14] and the difficulty in continuous manufacturing because of the blockage of the electrospinning nozzle by the conductive particles.

To overcome the above-mentioned problems, this study develops a conductive nonwoven material comprising cellulose nonwoven composite with carbon fibers (CNwCa) as the first layer, multiwalled carbon nanotubes (MWCNTs) as the second layer, and thermoplastic polyurethane (TPU) nanofibers as the third layer. The conductivity of the MWCNT composite conductive nonwoven fabric is high even if the MWCNT quantity is less because a binder is not used, and the carbon fiber and MWCNTs are connected to form a network. To achieve surface durability and conductivity, TPU with excellent physical properties is electrospun as the third layer to enable the microfiber pores to maintain the required level of conductivity.

In Levi’s^®^ Trucker jacket, developed by Google’s Jacquard Project [36], touch gestures such as swiping and tapping on the left cuff are used as commands. The jacket sensor is wirelessly connected to a mobile phone and photos can be captured, music can be played, and pausing, skipping, and navigation are possible through touch gestures. However, in order to execute the desired application, the mobile device application needs to be manually selected.

In this study, conductive nonwoven fibers were fabricated in a multilayered structure and showed excellent stability and durability during surface rubbing, folding, and washing. We also developed a mode-switchable touch sensor using conductive nonwoven fibers as sensors. This mode-switchable touch sensor can implement application selection mode, as well as gesture mode, so it has high potential in various fields such as wearable devices and outdoor electronics.

## 2. Materials and Methods

### 2.1. Materials

A cellulose nonwoven composite with carbon fibers (CNwCa) composed of 75 wt.% cellulose fibers and 25 wt.% carbon fibers was purchased from I-One film Co., Ltd., Anyang-si, Korea. Multiwalled carbon nanotubes (MWCNTs, purity > 95%) were purchased from ACN Tech, Pohang-si, Korea. Methanol (anhydrous, 99.8%) was purchased from Sigma-Aldrich Co., Ltd., Kenilworth, NJ, USA. CNT dispersant Disperbyk-2014 was purchased from BYK-Chemie GmbH., Wesel, Germany. Thermoplastic polyurethane (TPU, Elastollan 1195A10) was purchased from BASF Corp., Germany. *N*,*N*-Dimethylformamide (DMF) and tetrahydrofuran (THF) were purchased from Thermo Fisher Scientific, Waltham, MA, USA. Conductive nickel fabric tape (50 M EMI fabric tape, CFT-235) was purchased from Seyoung Tape, Seoul, Korea. All the materials were used as received without further treatment.

### 2.2. Preparation of the MWCNT Composite Conductive Nonwoven Fabric

#### 2.2.1. MWCNT Spraying Process

After dissolving 0.005 g of dispersant (Disperbyk-2014) in 50 mL of methanol, the dispersant solution was uniformly mixed through sonication for 20 min. The MWCNTs were evenly dispersed in the dispersant solution to obtain a final mass fraction of 0.03125 wt% and sonicated for 3 h. The MWCNTs were dispersed using an ultrasonic generator (ULH 700S, ULSSO HI-TECH Co., LTD, Korea) after which they were uniformly sprayed on the surface of the CNwCa substrate (10 cm × 10 cm) using an airbrush.

#### 2.2.2. TPU Electrospinning Process

TPU pellets were dissolved in a DCM/THF mixed solvent (ratio of 7/3, *v*/*v*) and stirred at room temperature. After stirring overnight, a viscous, colorless, and transparent polymer solution was obtained for electrospinning. Subsequently, 14 wt% TPU solution was added to a syringe with a metallic needle (inner diameter 0.8 mm) at an injection rate of 2.0 mL/h for electrospinning on the nonwoven MWCNTs. The loading voltage at the needle was 15 kV, and a grounded metallic roller wrapped in paper foil was used as the collector. The speed of the roller was set to 200 rpm for obtaining uniform nanofibers. The distance between the needle and collector was 22.5 cm. Electrospun fibers were fabricated under ordinary laboratory conditions, and the conductive nonwoven TPU microfiber was dried in a fume hood for 24 h. In Figure 1, the entire preparation process of the conductive nonwoven is illustrated.

### 2.3. Characterization

#### 2.3.1. Morphology of the MWCNT Composite Conductive Nonwoven

Field-emission scanning electron microscopy (FE-SEM, SU-8010, Hitachi Ltd., Tokyo, Japan) was performed after osmium coating the SEM specimens using a fine coater (JFC-1200, JEOL, Tokyo, Japan) for 30 s, for determining the morphology of each specimen.

#### 2.3.2. Morphology of the MWCNT Composite Conductive Nonwoven

Washing was performed using a laundry machine at 40 rpm, with a 5 g/L standard neutral detergent solution, for 100 min. As a wearable sensor can be exposed to sweat, moisture, and rain, the durability of the MWCNT composite conductive nonwoven fabric was analyzed by varying the resistance value before and after washing.

A smart wearable product with embedded sensors can be repetitively exposed to various physical impacts including artificial folding. To confirm the folding durability, the folding impact was repeatedly applied to the MWCNT composite conductive nonwoven fabric using an impacting machine [37] at a pressure of 15 kg_f_/cm^2^ and a cycle frequency of 0.5 Hz. The resistance value to static pressure was measured using a source meter under a load of 13.5 g_f_/cm^2^ with a standard weight on the sensor after 1000 cycles of folding. This process was repeated five times, and the value was measured.

### 2.4. Characterization

The developed mode-switchable touch sensor comprises a main circuit and sensing pad connected by wires, as shown in Figure 2a.

#### 2.4.1. Sensing Pad

The sensing pad includes four layers, as depicted in Figure 2b. The 1st and 4th layers function as electrodes. For the 1st layer, 20 mm wide nickel fabric tape was placed at intervals of 5 mm, and for the 4th layer, 10 mm nickel fabric tape was placed at intervals of 5 mm perpendicular to the 1st layer, as shown in Figure 2c. The developed MWCNT composite conductive nonwoven fabric, whose resistance changes with pressure, was used as the 2nd layer. A flexible printed circuit board (F-PCB) was used as the 3rd layer to connect the sensing pad and main circuit. The overall size of the touch sensor was 65 mm × 75 mm, and the sensing area was 50 × 60 mm.

#### 2.4.2. Main Circuit

The main circuit measures and processes the data of the sensing pad to control the mobile phone using Bluetooth. For this purpose, Nordic Semiconductor’s nRF52832 was used as the microcontroller. In the main circuit, a USB-C connector was used for the power supply, and a six-pin connector was used for connecting with the sensing pad. In addition, the main circuit with an overall size of 10 × 14 mm included driving circuitry for measuring the piezoresistive characteristics of the sensing pad, as shown in Figure 2d.

The schematic of the driving circuit is depicted in Figure 2e. A matrix structure was employed to measure eight points with six lines (two ADCs and four GPIO ports). To measure the resistance of each point, GPIO-0 alone was set to high (H) and the other GPIO ports were set as high impedance (Z), to ensure that the resistance of the other parts did not affect the measurement. The voltage was then measured at ADCs 0 and 1. The above process was repeated for GPIO- 0 to 4 in order to measure the voltage at all the points. Using the measured voltage, the resistance of the corresponding point was calculated according to Kirchhoff’s voltage law as follows:$$V_{out\left(i,j\right)}=\frac{R_{L}}{R_{L}+R_{s\left(i,j\right)}}A?V_{in},$$
where *i* is the GPIO number, and *j* is the ADC number. $R_{s\left(i,j\right)}$ is the resistance of each point, and $R_{L}$ is the load resistance (10 kΩ). $V_{in}$ is the H-state voltage of the GPIO port (3.3 V). $V_{out}$ is the voltage at each point measured by the ADC. The sampling frequency of the driving circuitry is 20 Hz, and the measured resistance classifies the gestures using a gesture classification algorithm.

## 3. Results and Discussion

### 3.1. Morphology of the MWCNT Composite Conductive Nonwoven

CNwCa used as the substrate of the micro nonwoven MWCNT composite contains 25% carbon fibers between cellulose fibers, as shown in Figure 3b. The CNwCa thickness is approximately 50 µm, and the carbon fiber diameter is approximately 16 µm. For imparting conductivity, MWCNTs were uniformly sprayed on CnwCa, with a diameter of approximately 50 nm and coating thickness of approximately 1–2 µm, as depicted in Figure 3c. To achieve surface durability and conductivity, 2–4 µm diameter TPU microfibers were electrospun with a coating thickness of approximately 5–7 µm. As shown in Figure 3d, the prepared conductive nonwoven fabric maintains conductivity by exposing the MWCNTs to the pores sufficiently formed between the TPU microfibers; in addition, durability is maintained since the pore size is considerably smaller than the touch size of a finger.

### 3.2. Optimization of the MWCNT Composite Conductive Nonwoven Fabrication Process

To optimize the electrical properties of the conductive nonwoven fabric before electrospinning, the content of the sprayed MWCNT was varied, and the connection effect of the CNwCa carbon fibers and MWCNTs was verified by comparing the difference in resistance values with and without CNwCa as a substrate.

As shown in Figure 4a, the resistance value is low, and the conductivity is improved when a CNwCa substrate is used because the carbon fibers and MWCNTs are connected to form a network. In addition, the optimum spray condition (MWCNT 5.2 wt.%) that realized suitable conductivity (268 Ω) and excellent uniformity (coefficient of variation (CV)% 6.1, Appendix A) was selected for the piezoresistive sensor.

With the increase in the electrospinning duration (0, 10, 15, and 20 min), the density of the TPU microfibers coated on the MWCNTs increase, and as a result, the resistance values increase to 268, 617, 986, and 1761 Ω, respectively (Figure 4b). This is because the formation of conductive paths through the skeleton is blocked by the high TPU content [38].

The pore size between fibers reduces (Figure 4c–e) and becomes increasingly uneven as more TPU microfibers are coated on the surface; hence, the CV% gradually increases to 6.1%, 6.3%, 7.4%, and 15.4%, respectively (Appendix A). In particular, the resistance value becomes significantly nonuniform after 20 min of electrospinning. The conductive nonwoven fabric to be applied to the sensor was sampled and used after 10 or 15 min of electrospinning considering the conductivity, uniformity, and durability.

### 3.3. Durability of the Conductive Nonwoven MWCNT Composite after Electrospinning

As the conductive layer of conventional conductive textiles manufactured using MWCNTs lacks durability, the surface cracks easily due to external forces. Using a considerable quantity of binder to solve this problem is disadvantageous because the layer becomes stiff and thick and the conductivity is significantly lowered.

In this study, to effectively impart conductivity with a small quantity of MWCNTs, CNwCa-containing carbon fibers were used as the substrate on which MWCNTs were sprayed without a binder. After spraying, the surface with MWCNTs was coated using an electrospinning process that facilitated micropore formation, maintaining adequate conductivity and durability.

Figure 5a,b intuitively shows that the conductive nonwoven manufacturing method proposed in this study increases the surface durability. This method is expected to be economically and widely commercialized because it is simple and allows continuous processing.

In addition, the results of the washing test for 100 min (Figure 5c) establish the moisture durability of the developed conductive nonwoven fabric. The resistance was measured 50 times before and after washing. The resistance value and CV% increase slightly after washing, but there is no significant difference, confirming that the TPU microfiber layer imparts water resistance to the conductive nonwoven fabric.

In the repeated folding experiment (5000 times), the change in the resistance value and CV% are insignificant up to 3000 times and a slight change occurs beyond 4000 times, as depicted in Figure 5d. After repeatedly folding 5000 times, the resistance value increases by 4.7%, and the CV% is above 10%; however, considering the severe test folding conditions (15 kg_f_/cm^2^), the developed conductive nonwoven fabric is sufficient for use as a wearable sensor.

### 3.4. Piezoresistive Properties of the MWCNT Composite Conductive Nonwoven

The developed conductive nonwoven fabric exhibits piezoresistive properties in which the resistance decreases with the increase in load. The measured resistance is in the range of 1091–125 Ω when the applied pressure is varied from 4 to 88 g_f_/cm^2^, and the highest CV% of the measured resistance is 12.2 (Figure 6, Appendix A). The obtained results establish that the conductive nonwoven MWCNT composite can be applied to pressure sensors that can sense even 4 g_f_/cm^2^. High correlation is indicated by the value (0.97) of the coefficient of determination (R2) of the correlation equation (X-load, Y-calculated resistance). Thus, the load can be predicted with very high accuracy based on the resistance value measured by the sensor.

### 3.5. Applications of the Touch Sensor Using Conductive Nonwoven MWCNT Composite

#### 3.5.1. Airbag Touch Sensor for Soft Robot

As the fabricated touch sensor has excellent flexibility, it can be applied to curved surfaces and elastic materials. Hence, it was applied as the airbag material used as the outer skin of a soft robot arm. As shown in Figure 7, the touch sensor fabricated in this study can detect various motions, such as slide, double-tap, stroke, and cover (Appendix A).

#### 3.5.2. Smart Bag Prototype with a Mode Switchable Touch Sensor

In this study, a smart bag embedded with the fabricated touch sensor was developed. The smart bag was connected to a mobile device, enabling the convenient execution of the various functions of the mobile device through simple gestures while moving or in action.

Although conventional textile-based touch sensors applied to clothing or bags can recognize various gestures (slide, tap, cover, etc.), mode switching is difficult. However, the mode-switchable touch sensor developed in this study can switch modes through a simple gesture and also recognize various gestures in each mode.

Five modes (main, camera, music, document, voice command) were implemented in the smart bag prototype in which four modes (camera, music, document, voice command) can be selected through a tap gesture in the main mode, and each mode can be switched to the main mode through cover gestures. The function of each gesture is depicted in Figure 8 for each mode.

The type of gesture was recognized using the ADC value measured every 50 ms in the eight sensing cells of the mode-switchable touch sensor. The gesture determination algorithm (Figure 9) is as follows:If the ADC values of all the sensing cells are less than 30, it is not recognized as a gesture;If the maximum ADC value of all the sensing cells is greater than or equal to 30, it is determined that the corresponding cell has been touched;If there is a touched cell, and the sum of the ADC values of all the cells is greater than 200, and the standard deviation is less than two, it is recognized as a cover gesture;If there is a touched cell in the current sampling, and there is no touched cell in the next sampling, it is recognized as a tap gesture;If there is a touched cell in the current sampling and next sampling, and the touched cells in the current sampling and next sampling are different, it is recognized as a slide gesture;If there is a touched cell in the current sampling and next sampling, and the touched cells in the current sampling and next sampling are the same, it is recognized as a long tap gesture.

The mode-switchable touch sensor developed in this study can perform various functions without requiring the removal of the mobile device from the bag; hence, it is expected to be highly beneficial during outdoor activities such as hiking and trekking. As an example of this, Figure 10 shows the operation of a prototype bag to which a mode switchable touch sensor is applied.

## 4. Conclusions

A high-performance conductive nonwoven fabric was fabricated by depositing MWCNTs on cellulose nonwoven composites with carbon fibers (CNwCa) through a spray process to assign conductivity, and this was followed by the electrospinning of TPU on the MWCNT-coated CNwCa to improve surface durability. A uniform resistance value of 600 Ω suitable for a touch sensor was realized by controlling the conditions of the MWCNT spray process and the electrospinning process. It was established that the physical durability was stable with variations in the resistance value and a CV% of approximately 5% was determined in repeated folding experiments (repeated more than 3000 times). In addition, a prototype bag with a touch sensor was fabricated using the developed conductive nonwoven fabric, and the bag’s touchpad function was demonstrated using an Android application.

The developed touch sensor measures the pressure change in various motions and accurately performs the input function and also maintains its function despite repeated physical damage with excellent durability. Based on the advantages of excellent durability, various functions, and flexibility, the touch sensor of this study can be widely used in home care, pressure sensing mats, as well as wearable devices. In particular, the mode switching function can perform various functions without removing the mobile device from the bag, and therefore, it is expected to be highly advantageous during outdoor activities such as hiking and trekking.

## Figures and Tables

**Figure 1 polymers-14-01545-f001:**
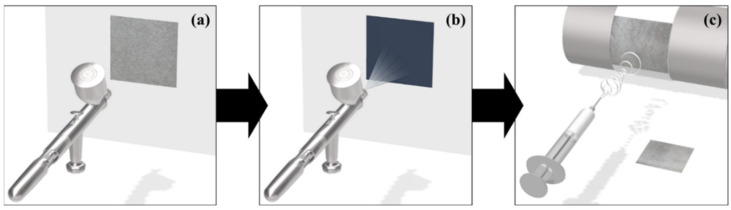
Illustration of the MWCNT composite conductive nonwoven preparation: (**a**) CNwCa, (**b**) MWCNT spraying process, and (**c**) electrospinning process.

**Figure 2 polymers-14-01545-f002:**
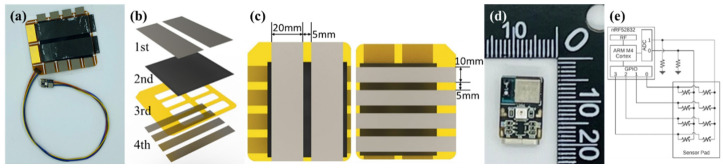
Mode-switchable touch sensor: (**a**) whole image (**b**) sensing pad layer composition, (**c**) dimensions, (**d**) main circuit, and (**e**) schematic of the driving circuit.

**Figure 3 polymers-14-01545-f003:**
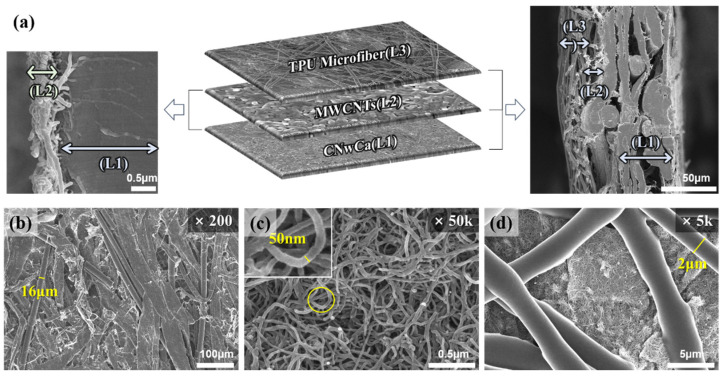
(**a**) Scanning electron microscope (SEM) images and configuration of the MWCNT composite conductive nonwoven. SEM images of (**b**) CNwCa, (**c**) MWCNTs on CNwCa, and (**d**) TPU microfibers on MWCNTs + CNwCa.

**Figure 4 polymers-14-01545-f004:**
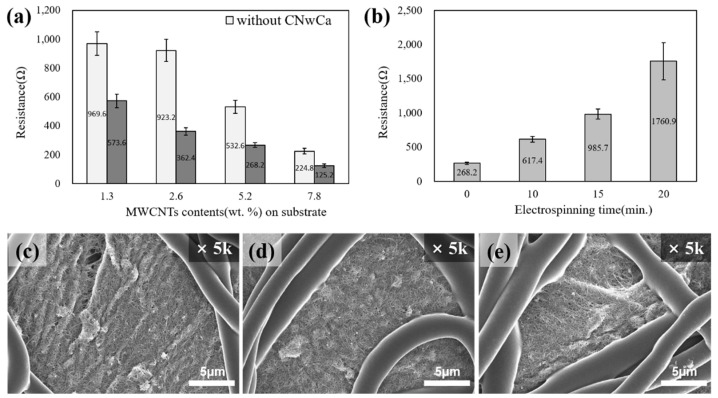
Resistance of the conductive nonwoven MWCNT composite after the spraying process with respect to the (**a**) MWCNT content and (**b**) electrospinning duration. SEM images at electrospinning duration of (**c**) 10 min, (**d**) 15 min, and (**e**) 20 min.

**Figure 5 polymers-14-01545-f005:**
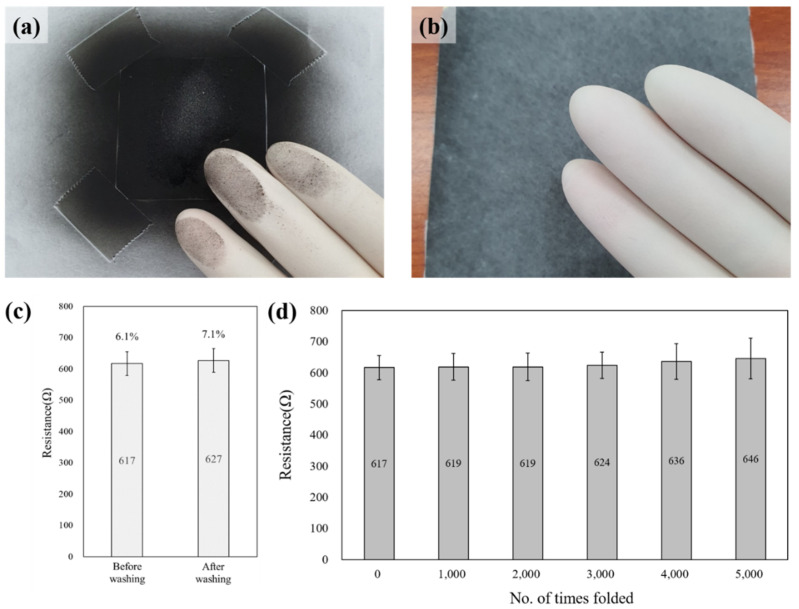
Surface durability (**a**) before and (**b**) after electrospinning of the conductive nonwoven MWCNT composite. Resistance value of the MWCNT composite conductive nonwoven (**c**) before and after washing and (**d**) with respect to the number of times folded (every 1000).

**Figure 6 polymers-14-01545-f006:**
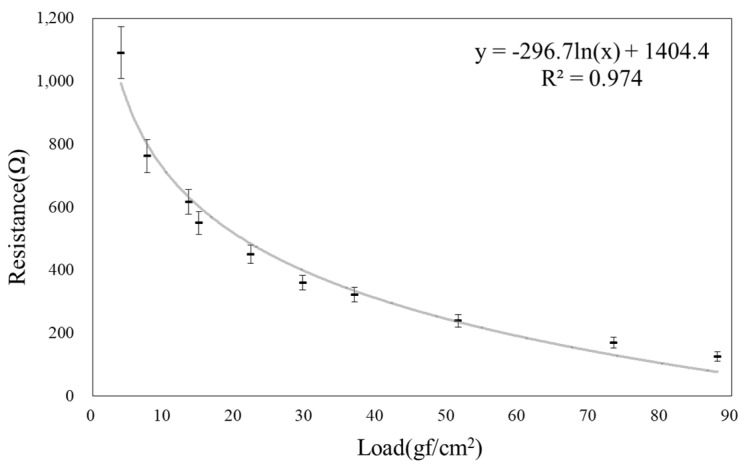
Load–resistance calibration curve of the conductive nonwoven MWCNT composite.

**Figure 7 polymers-14-01545-f007:**
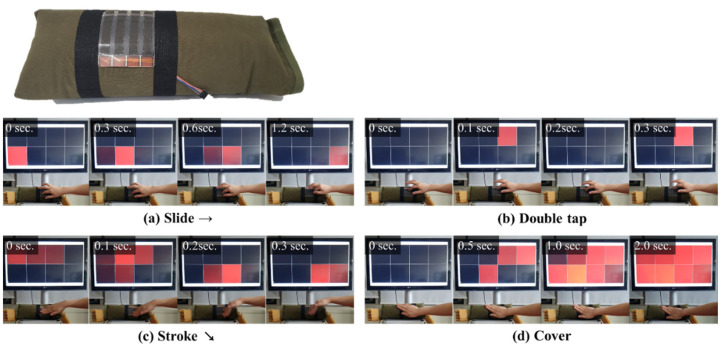
Touch sensing images of the airbag skin prototype at various dynamic pressure.

**Figure 8 polymers-14-01545-f008:**
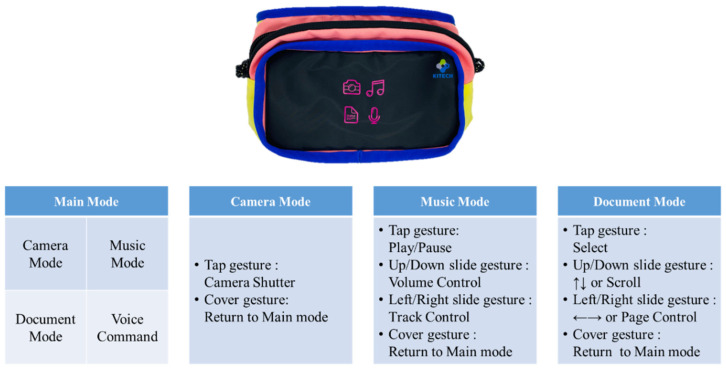
Prototype bag including a touchpad with conductive nonwoven fabric and function of each gesture in each mode.

**Figure 9 polymers-14-01545-f009:**
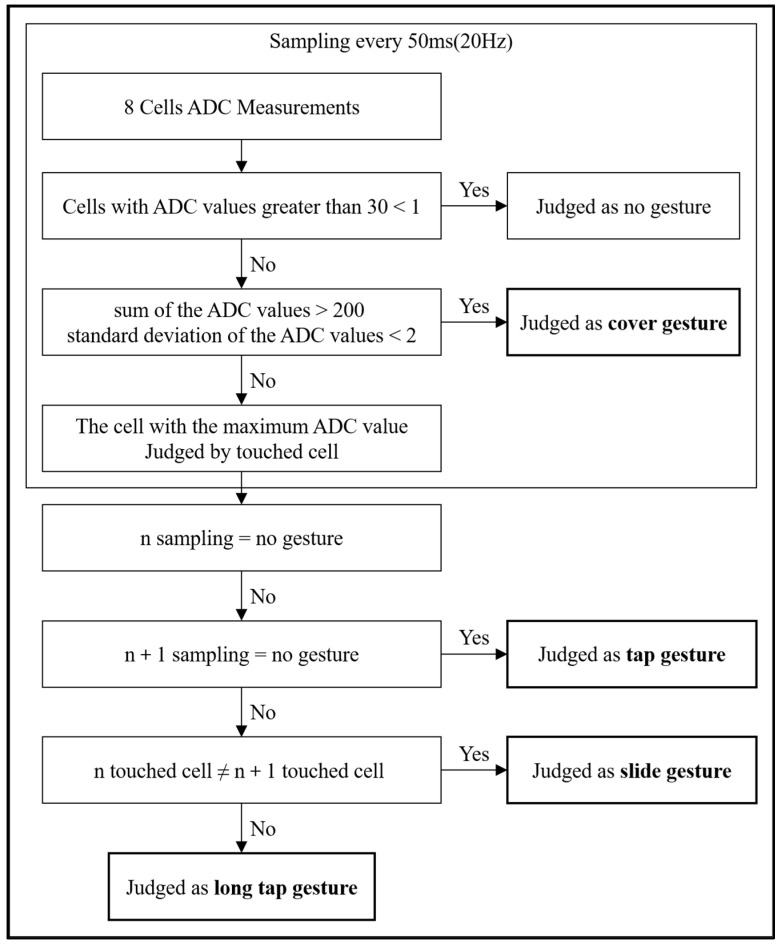
Examples of the operation of the camera, music, document, voice command, and mode conversion of the developed smart bag prototype (Appendix A).

**Figure 10 polymers-14-01545-f010:**
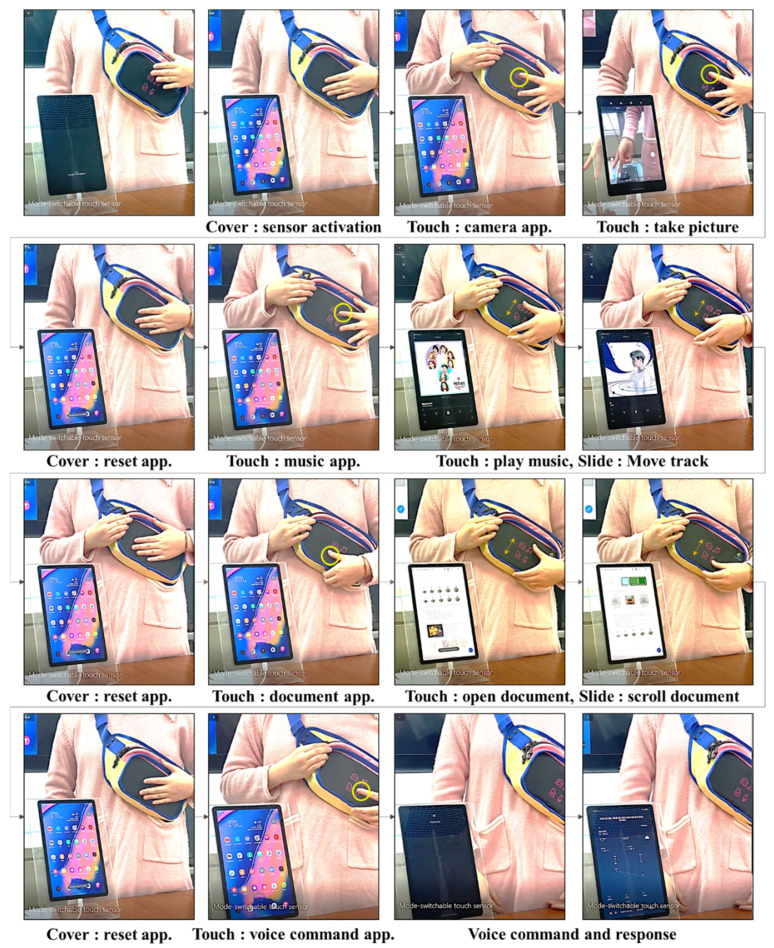
Examples of the operation of the bag prototype with a mode switchable touch sensor.

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
