# Peer review of "Development of Mode-Switchable Touch Sensor Using MWCNT Composite Conductive Nonwoven Fabric"

_polymers, 2022, doi:10.3390/polym14081545_

Round 1
Reviewer 1 Report
Jang et al have demonstrated fabrication of conductive nonwoven fabric using MWCNTs on cellulose-carbon composite followed by electrospinning thermoplastic polyurethane. A protype bag with touch sensor is demonstrated here using conductive cellulose nonwoven fabric. I recommend for publication after addressing following comments.
Comments to be addressed.
- There are lot of typing error throughout the manuscript. On page 4, line 171, Page 5, line 173, 174, 175, etc.
- In SEM micrograph, the scale is not visible. I will help reader for better understanding.
- At couple place, the discussion is not clear. Page 5, line 178-179 and Page 6 line 235-237 is not clear.
- Supporting Table 1 to Table 5 are missing is the manuscript.
- Effect of multiple washing cycles on conductivity should be addressed to evaluate the durability of the conductive nonwoven fabric.
- Mechanical testing will be helpful to determine the max strain before failure, conductivity loss, etc.
Reviewer 2 Report
The manuscript titled “Development of mode-switchable touch sensor using MWCNT composite conductive nonwoven fabric”, reports the fabrication by depositing multiwalled carbon nanotubes on cellulose nonwoven composites with carbon fibers through a spray process to assign conductivity, followed by electrospinning thermoplastic polyurethane on this nanocomposite to improve surface durability. In addition, the development of a prototype tactile sensor bag using this nanocomposite is shown and its touchpad function is experimentally verified by placing the sensor in a bag and using an Android application. Therefore, the present manuscript shows some novelties and I think that the manuscript is suitable for publication in Polymers, after careful and major revisions. Here are some suggestions for the revised version:
- In the introduction, I advise the authors to summarize in a few lines the important properties shown by carbon nanotubes, graphene, and carbon nanoparticles. I suggest adding the following notes: Journal of Industrial and Engineering Chemistry 2014, 20, 1171–1185; Nanomaterials 2020, 10, 2549.
- The authors should be clearer when talking about the MWCNT quantities used to obtain a final mass fraction of 0.03125wt%. Also, how did they weigh 50 g of methanol? They should be more scientific.
- The four layers must be indicated with the corresponding numbers in Figure 2b. The description of the sensing pad is unclear and confusing. For example, in the sentence “As the 1st layer, 20-mm wide nickel fabric tape was placed at intervals of 25 mm, and as the 4th layer, 10-mm nickel fabric tape was placed at intervals of 15-mm perpendicular to the 1st layer.” What does it mean intervals of 25 or 15 mm? How were the Ni cloth electrodes prepared? What is the main circuit?
- The authors should show the chemical map by EDX of the systems investigated with the SEM (Figure 3).
- Authors should carefully review the text, which is filled with typos, incorrect unit formatting, and sentence repetitions. Important editing is required.
- For example, kgf/cm2 (correct Kgf/cm2), 20-mm (correct 20 mm), etc. In the paragraph “Morphology of the MWCNT composite conductive nonwoven” the units of thickness and diameter are not included (lines 171, 173, 174, 175).
- line 179 what does “iop [xbnm, q6i03” mean?
- lines 235-237 delete the sentence “Figure 3. Resistance of the conductive nonwoven MWCNT composite after the spraying process with respect to the (a) MWCNT content and (b) electrospinning duration. SEM images at electrospinning duration of (c) 10 min, (d) 15 min, and (e) 20 min.” repeated in Figure 5.
- In the introduction, it is necessary to add the main fields of use of both carbon nanotubes, graphene, and carbon nanoparticles. I suggest adding the following notes: Polymers 2021, 13, 3190; Chemistry-A European Journal 2021, 27, 13715–13718; ACS Appl. Nano Mater. 2020, 3.8182−8191; Polymers 202
Round 2
Reviewer 2 Report
I thank the authors for satisfactorily answering my questions. The review is now better than it was in the beginning.
Due to the excessive number of characters the two notes that authors should add in the introduction application are: Polymers 2022, 14, 542; Molecules 2020, 25, 5731.
